# Unveiling the Paradox of NFT Prosperity

## ABSTRACT

Unlike fungible tokens (e.g., cryptocurrency), a Non-Fungible Token (NFT) is unique and indivisible. As such, they can be used to authenticate ownership of digital assets (e.g., a photo) in a decentralized fashion. Given that NFTs have generated significant media attention since 2021, we perform a large-scale measurement study of the NFT ecosystem. We collect over 242M transfer logs and over 97M marketplace transactions until Aug 1st, 2023, by far the largest NFT dataset, to the best of our knowledge. We characterize the on-chain behavior of NFTs and their trading across five major marketplaces. We find that, although the NFT ecosystem is growing rapidly, it is driven by a relatively small set of dominant centralized players, with suspicious trades activities, e.g., over 23% of the monetary volume is generated by malicious wash trading and the ecosystem has experienced over 157K cases of NFT arbitrage, with a total sum of over $25M USD profit. Our observations motivate the need for more research efforts in the NFT security analysis.

## 1 INTRODUCTION

There has been significant media and market attention surrounding Non-Fungible Tokens (NFTs) [1, 11]. These are a kind of cryptographic token that is *unique* and *indivisible*. Each NFT is *one-of-a-kind* and can be used to authenticate ownership of a single digital entity, e.g., a photo. As all exchanges of NFTs are recorded on a blockchain, they can be used to prove the ownership of a particular asset. This simple concept has spurred interest, assisting users to trade non-fungible goods in a decentralized fashion. Yet, many are concerned about the economic risks of NFTs, as their rapid growth [8] has attracted various anecdotal fraudulent attacks.

Although there has been recent work [42] on NFTs *themselves*, we lack answers to (even basic) questions that are associated with *NFT markets*, such as (*i*) How can we systematically collect data from NFT markets? (*ii*) How often are NFTs traded and for what price? (*iii*) Which are the most dominant marketplaces and what role do they play in underpinning the wider ecosystem? (*iv*) Are NFTs subject to price fraud, or other associated types of market manipulation?

To explore these issues, we conduct a comprehensive study of the NFT market ecosystem. Our focus is on the digital tokens themselves (NFTs) and the platforms where people buy and sell them. *First*, we aim to examine the growth of the NFT ecosystem, which includes tracking NFT-related events, the number of participants involved, and how these marketplaces operate — particularly if there are any unfair practices. *Second*, we aim to explore the possibility for market manipulation within the ecosystem. Based on anecdotal reports [14, 21], we strive to systematically understand the severity of this problem.

To achieve these aims, we collect over 242M transfer logs and 97M marketplace trades until Aug 1st, 2023 (§3). After that, we conduct a graph analysis of NFTs, as well as how they are exchanged via NFT marketplaces (§4). We identify preliminary evidence of potential market manipulation, and this inspires us to perform a

rigorous analysis of two specific cases (§5): (*i*) *wash trading*, where users repeatedly exchange NFTs between accounts they control to simulate artificial demand; and (*ii*) *arbitrage*, where users strategically sell and buy across marketplaces to exploit fluctuations in price. We find that both are commonplace, with worrying implications: over 23% of the NFT market's monetary volume is fake (generated artificially by wash trading). This raises serious concerns over the sustainability of the NFT market.

We make the following research contributions in this paper:

- *We perform a large-scale graph analysis of the NFT ecosystem.* We gather a dataset covering over 24M NFT smart contracts, 142M NFTs, 242M transfer events and 97M trade events. We expose a growing ecosystem, driven by a relatively small set of dominant players with unhealthy behaviors.
- *We measure the prevalence of wash trading behavior in the NFT ecosystem.* We reveal that NFTs experience significant price manipulation by at least 826 wash trading bots. In total, these bots account for at least over $24B USD of history volume growth (over 23%) in the NFT ecosystem.
- *We propose a methodology to detect the arbitrage of NFTs.* Our proposed detection method reveals that over 157K instances of NFT arbitrage exist in the wild, with the profits of over $25M USD conducted by 629 accounts. All datasets will be made publicly available.

We will release our results to the research community.

## 2 BACKGROUND

### 2.1 Ethereum Primer

**Ethereum.** *Ethereum* is one of the most popular blockchains. Its key innovation was the introduction of *smart contracts*, and it is the de-facto technology used for NTFs. *Ether (ETH)* is the native cryptocurrency on *Ethereum*, the second largest cryptocurrency after *Bitcoin* [3].

**Ethereum Account.** *Ethereum accounts* are identified by a fixed-length hash-like address, which can be divided into *external-owned accounts (EOAs)* and *contract-owned accounts (COAs)*. *EOAs* are controlled by users, i.e., anyone with private keys, while the *COAs* are controlled by code stored together with the accounts. An *EOA* is an ordinary account that can transfer tokens, invoke deployed *smart contracts* and store received tokens. Moreover, an *EOA* can deploy a *smart contract* into a *COA* and a *COA* can only send *transactions* in response to receiving *transactions*.

**Ethereum Transactions.** When a user wants to interact with *Ethereum*, a *transaction* is made through their *EOA* to modify or update the state stored in *Ethereum*.

**Etherum Smart Contracts.** A *smart contract* consists of code that implements actions using *transactions*. Based on the foundation of *smart contracts*, *ERCs* (Ethereum Request for Comments) have proposed a series of standards for digital tokens in *Ethereum*.

## 2.2 Digital Token and DeFi

**Tokens.** Each token belongs to a *token smart contract*, which defines a set of functions used to perform different tasks. One prominent example is *ERC-20*, which is non-unique and divisible [6]. In a token smart contract under the *ERC-20 standard*, all tokens are the same and have the same value.

**NFTs.** A *Non-Fungible Token (NFT)* is a kind of cryptographic asset implemented on a blockchain. *NFTs* are used to identify content in a digital way. Such content includes paintings, videos or other items in the real world. The ownership of the NFT is recorded via a *transaction* on the blockchain. Thus, theoretically people can verify the ownership.

**ERC-721 and ERC-1155.** *ERC-721* defines a minimum set of interfaces which a smart contract must implement to manipulate the NFT tokens on *Ethereum*. Each ERC-721 NFT has unique *ID* and identifies one unique piece of content, which means they cannot be divided into smaller units. However, when we need many different kinds of *NFTs* to operate, using *ERC-721* is inefficient since it needs to create many *ERC-721* contracts. To address this, *ERC-1155* was proposed to manage multiple token types in a single *smart contract*. The unique *ID* of a *ERC-1155* smart contract points to a batch of tokens that have the same content. If someone needs to transfer a batch of tokens, they can execute a single *transaction* (rather than multiple ones), which consumes less *gas* (the fee required to conduct a transaction or execute a contract).

**Decentralized Exchanges.** *Decentralized exchanges (DEXes)* provide peer-to-peer marketplaces for investors who want to trade digital tokens. The *DEXes* have their own smart contracts launched to deal with the events the transactions generate through DEXes.

**NFT Secondary Marketplaces.** In the NFT ecosystem, the NFT exchanges (aka "*secondary marketplaces*") play the role of *DEXes*. Five top platforms dominate the NFT market: *OpenSea* [10], *LooksRare* [13], *CryptoPunks* [4], *LooksRare* [9], and *Blur* [2]. They each have their own unique official smart contracts that have been launched on *Ethereum*. They also have front-end websites which provide a convenient place for NFT trading.

## 2.3 The Life Cycle of an NFT

**NFT Creation.** An NFT smart contract (which normally implements either *ERC-721* or *ERC-1155* tokens) implements all features and functions of one NFT project. After the launch, other participants can perform the "mint" function to create an NFT. Normally, the qualification of minting tokens is sold to the public as a chance to be added to the *whitelist* of the projects' smart contract. The accounts then have the privilege to perform the mint operation and generate a *mint event*, as well as to gain authority over the token. Note, NFT smart contracts on *Ethereum* have an "approve" operation which allows users to grant their privileges on tokens to other accounts. Note that, NFT can also be burned, i.e., destroying it by sending an NFT to an un-spendable address.

**NFT Trading.** NFTs rely on a *secondary marketplace* for circulation, where token owners can list their NFTs. In a marketplace, the NFTs of a project always appear as a "collection", which is an off-chain concept and can be seen as "brands" in the NFT world. Normally, one smart contract maps to one collection. Optionally, sellers can list their NFTs on multiple marketplaces and users can place *bids*

**Table 1: Dataset overview.**

| Data Type | # Number | Type | # of Transfer Events | marketplace | # of Trade Events |
|---|---|---|---|---|---|
| Smart Contract | 244,154 | Mint | 148,500,667 | OpenSea | 93,128,954 |
| Token ( Except ERC-1155 ) | 142,561,997 | Burn | 917,025 | X2Y2 | 2,264,694 |
| Transfer Event | 242,444,962 | Swap | 93,027,270 | CryptoPunks | 30,839 |
| Trade Event | 97,902,053 | | | LooksRare | 620,789 |
| | | | | Blur | 1,856,777 |

on them. When an *offer* is accepted, the website will automatically invoke their official smart contract to deal with this event, and generate a *swap* event. For full details, we redirect readers to [35].

## 3 DATASETS

**Token Transfer Dataset.** We use *Geth* [16] to download the *Ethereum* ledger. We first synchronize all blocks until Aug. 1st, 2023. We then extract four parts of data from these blocks: *external transactions*, *internal transactions*, *contract information*, and contract calling information. We then trace every NFT contract and extract other information directly from the blockchain. We extract all 242, 444, 962 transfer events.

**NFT Secondary Market Trade Dataset.** We next compile data covering the trades that take place within marketplaces. Note, a trade is different to a transfer: a *trade* takes place within the smart contract of a secondary marketplace (for a sum of money), whereas a *transfer* is the event that transfers NFT ownership to another account on the first market (i.e. the Ethereum). We start by manually analyzing the smart contracts of five major NFT markets to see how they execute NFT trades: *OpenSea*, *X2Y2*, *CrypotoPunks*, *LooksRare* and *Blur*. These cover over 98.1% of the total trade volume in Ethereum [5]. The specific contract analysis and data collection methods are detailed in Appendix A. We gather 97,902,053 data items in our NFT secondary market trade dataset until Aug. 1st, 2023.

**NFT Smart Contracts and NFTs Dataset.** To identify all NFT smart contracts and tokens, we simply extract all the *ERC-721* and *ERC-1155* token's transfer events in the external transaction logs. In total, we identify over 244,154 NFT smart contracts. Note, because smart contracts under the *ERC-1155* standard could be called to mint a huge number of tokens at one time, it is meaningless to count the *ERC-1155* tokens. While minting a token, a specific transfer event is generated (on the blockchain) whose transfer from is the *null address*. Thus, we count this type of transfer event and filter out *ERC-1155* transfer events to calculate the number of NFTs. This gives us 142,561,997 NFT tokens in total. To the best of our knowledge, this is the most complete dataset of NFTs available.

**Dataset Overview.** Table 1 summarizes the data we have collected, consisting of the data type, transfer type and trade marketplace. In total, we have collected over 244,154 NFT smart contracts, 128M NFTs, 242,444,962 transfer events and 97,902,053 marketplace trade events. For analysis, we further divide the transfer events into three types. For those transfer events whose "transfer from address" is the null address, we label them as *mint events*. For those whose "transfer to address" is the burn account [15], we label them as *burn events*. This is where the user removes the tokens from the overall supply (aka "burning"). For the remaining tokens, we label them as *swap events*, whereby an NFT is transferred to another owner.

# 4  NFT ECOSYSTEM DEVELOPMENT

## 4.1  Exploration of NFTs Events

We first inspect the activity and usage of NFTs by dissecting the various NFT events recorded.

**Mint Events of NFTs.** A *mint event* is when a smart contract is used to create a new NFT. Fig. 1(a) presents a time series of the number of *ERC-721* and *ERC-1155* tokens minted. When the *ERC-721* standard was first proposed in 2018, it did not attract much attention. But since the beginning of 2021, the creation of *ERC-721* tokens has become far more frequent, with significant growth. This is primarily driven by the growing use cases of NFTs. The total number of *mint events* of ERC-721 smart contracts in Jan 2021 hit 96,771, while the number is 4,518,268 in Jan 2022, which has increased over 46 times.

There have also been serious fluctuations during this period. For example, from the middle of Sept 2021, the daily creation rate dropped rapidly, before rebounding again in 2022. Overall, the rate of *ERC-721* tokens creation has been higher than that of *ERC-1155* tokens. Closer inspection further reveals significant peaks. For example, from Oct. 29th, 2019, to Nov. 18th, 2019, the number of mints per day is above $10^5$, where it reaches a peak on 2019.11.17 (with over 4.8M mints). We find that the project *Gods Unchained Cards* performs the majority of minting during that period (a digital trading card game). During this period, it minted many cards to satisfy the needs of its players. This phenomena highlights that the behavior of the overall ecosystem can be heavily affected by a single (non malicious) influential smart contract.

We also inspect the distribution of *mint events* across all NFT contracts. Fig. 1(b) and (c) present the number of mint events per contract for *ERC-721* and *ERC-1155* contracts, respectively. 23.1% of *ERC-721* smart contracts only mint one token, and 49.1% of the smart contracts mint no more than 5 tokens (64.4% mint no more than 20 tokens). The characteristics of *ERC-721* contracts are similar with *ERC-1155* contracts, although overall *ERC-1155* contracts tend to mint more tokens. The respective percentages for *ERC-1155* are 35.9%, 61.1%, and 74.8%. Thus, a small number of smart contracts mint the majority of NFTs: The top 10% of contracts mint 90.57% of all tokens. This raises serious concerns about the the true level of decentralization in the ecosystem, as the removal of a small number of stakeholders would remove the majority of "creativity".

**Swap Events of NFTs.** To explore how active these tokens are, we next look the number of swap events for each token. Recall, a *swap* event is where the ownership of an NFT is transferred to another. Fig. 2(a) presents a time series distribution of the number of token *swap events*. We see that *swap events* became frequent in the beginning of 2021 and have grown by 5581% since (Jan 2021 – Sept 2022). Much like the token mint timeline, the curve fluctuates heavily and the swap rate of *ERC-1155* tokens is less than *ERC-721* tokens for the same reason discussed above.

Fig. 2(b) and Fig. 2(c) present the distribution of *swap events* per contract for *ERC-721* and *ERC-1155* contracts, respectively. We observe a large range among the number of *swap events*. Whereas most tokens are transferred a small number of times, we observe an elite that experience extremely heavy circulation. Only the top 1% have been transferred over 20 times. Consequently, we observe a long-tail of highly undesirable NFTs. 73.1% of *ERC-721* NFTs have never been transferred (77.2% for *ERC-1155* tokens); and 98.9% (98.4%) of them have fewer than 5 swap events. This suggests that the majority of NFTs are rather undesirable and experience little market activity.

**Burn Events of NFTs.** Finally, we inspect the number of *burn events* for NFTs. A *burn event* is where an NFT is deleted from the supply. As shown in Table 1, we identify 917,025 *burn events*. There are only 12,652 (4.96% of the total) smart contracts that have one or more burn events. This is perhaps surprising as it is not clear why one would "burn" an NFT. To understand the reasons, we manually investigate 100 NFT projects that have *burn events*, and observe the following reasons. First, some projects burn for corner-case reasons. To highlight this we take the example of the *OpenSea Shared Storefront* smart contract, which has the huge number of burn events (33,982). It is the official contract from *OpenSea*, an NFT marketplace: It does not only support one collection, but many (in fact, it allows users to mint their own NFTs). Thus, the contract burns NFTs that are removed from the market, e.g., because they are reported to be scams. Second, *ERC-1155* NFT projects appear to burn their NFT tokens to reduce the total supply. For example, we check the *ERC-1155* NFT project *PAGE* [17] that has the second largest number of *burn events* (29,045). Unlike *ERC-721* tokens, the *contract address* and *token ID* belong to a set of tokens with the same price. In this case the *ERC-1155* tokens are therefore practically the same as traditional cryptocurrency tokens (i.e., ERC-20 tokens). Burning them can therefore reduce supply, thereby increasing their price. Third, since there are many NFTs airdropped to other accounts like *spam emails*, *EOAs* also burn the tokens by themselves, to avoid accidentally clicking on a fraudulent link.

## 4.2  Exploration of Participants

We next explore *who* drives the above NFT events (i.e., the accounts). We first define a weighted directed graph, the *transfer account graph*, i.e., $TAG = (V, E, w)$, where $V$ is a set of accounts, $E$ is a set of edges, and $w$ is a set of integers indicating the number of transfers between two different accounts. There are 8,189,043 nodes (i.e., accounts in the NFT ecosystem) with 242,444,962 edges (i.e. transfer events). Note, we include the "null" account from which all new NFTs are initially transferred. To generalize this, Fig. 3(a) and (b) show the in and out degree distributions. As expected, the distributions are highly skewed. As with prior analysis, we observe a long tail — 40.8% of accounts have an in-degree of 1, with 35.4%, having an out-degree of 1. Just 12.8% have an in-degree over 20 (85.8% for out-degree). This suggests significant centralization in the production of NFTs.

To better understand these influential accounts, Table 4 and Table 5 of the Appendix B list the top five accounts, as measured by in and out-degree. In total, these five accounts cover 3.06% of in-degree and 64.94% of out-degree, respectively. The discrepancy is because the mint events generate a transfer events whose "from address" is null (see §3). Thus, the null address has an out degree of 148,500,667 (61.25% of the total out degree), which reveals *the low liquidity of NFTs*. Beyond the null account, we further conjecture that other accounts with very high degrees might be automated. By searching these accounts, we observe a number of automated services (see Tables 4 and 5 of the Appendix). For example, the *Ethereum*



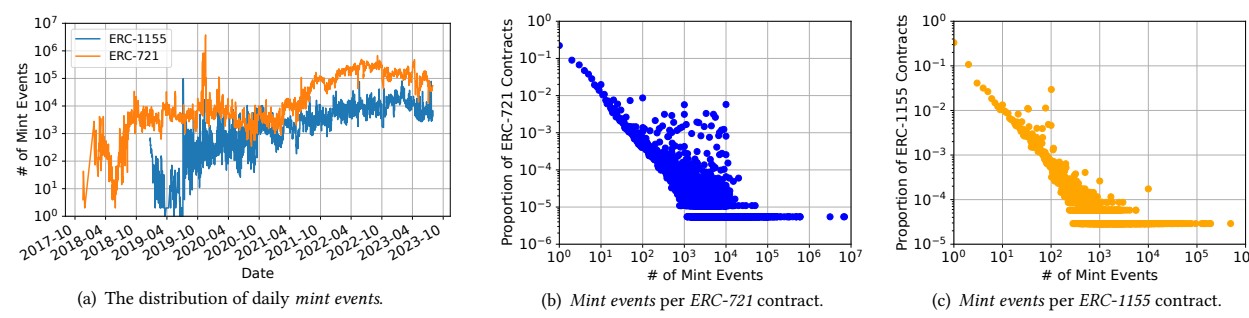

(a) The distribution of daily *mint events*.  (b) *Mint events* per *ERC-721* contract.  (c) *Mint events* per *ERC-1155* contract.

**Figure 1: Graphs of *ERC-721* and *ERC-1155* of *mint events*.**

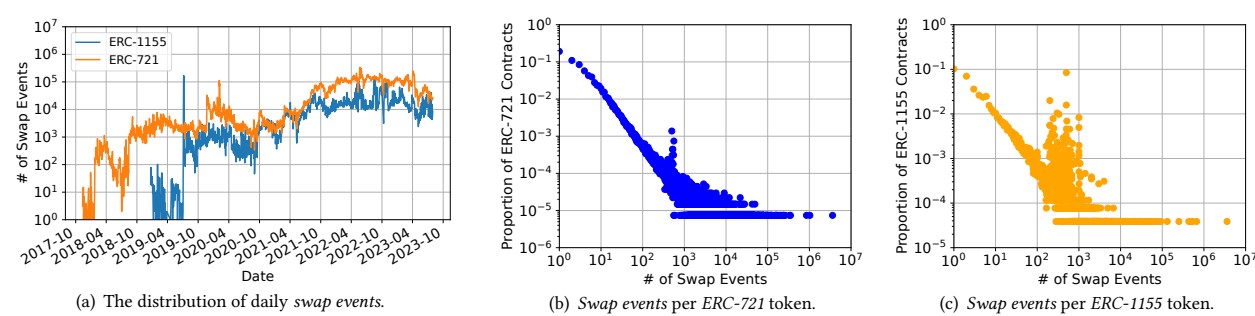

(a) The distribution of daily *swap events*.  (b) *Swap events* per *ERC-721* token.  (c) *Swap events* per *ERC-1155* token.

**Figure 2: Graphs of *ERC-721* and *ERC-1155* of *swap events*.**

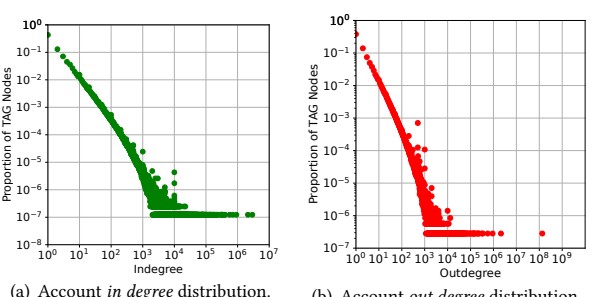

(a) Account *in degree* distribution.  (b) Account *out degree* distribution.

**Figure 3: Overview of the *transfer account graph* (TAG).**

*Name Service (ENS)* is a naming system based on the Ethereum blockchain, which maps human-readable names (e.g., *alice.eth*) to machine-readable identifiers. Another example is *MetaWin*, which aims to provide a community-oriented brand by investing in opportunities centered around NFTs. These accounts are Dapps on Ethereum, providing different function to the NFT ecosystem. Importantly, we do not see *any* personal trading accounts attaining this large number of transfer events.

## 4.3 Exploration of Marketplaces

**Marketplaces Overview.** Recall that, the marketplaces we measure (*OpenSea*, *X2Y2*, *CrypotoPunks*, *LooksRare* and *Blur*) cover 98.1% of total trade volume in *Ethereum* in 2022 [5]. Fig. 4 presents their number of users, cumulative NFT price (i.e., volume), and transactions.

*OpenSea* is the most successful marketplace (across all three metrics). *OpenSea* and *CryptoPunks* are the longest running NFT marketplaces. *LooksRare* and *X2Y2* were launched later in 2022, but also have stable daily users, transactions and a large price volume. However, they are collapsing now. *Blur*, as a new market, has significant growth in 2023. After NFTs became popular in 2021, the sum price within *CryptoPunks* rapidly increased in value and held a high daily cumulative price volume (almost higher than *OpenSea*), yet only had an average of just 1,654 transactions and 1,924 users. This surprising observation is explained by the nature of the *CryptoPunks* marketplace. It was launched in 2017 with 10,000-pixel images, also called "The first non-fungible token" [34]. This small set of NFTs gained significant attention, resulting in high price trades amongst a small number of individuals. *LooksRare* has far fewer transactions on average, but occasionally outstrips *OpenSea*, with around $10^3$ daily transactions and $10^3$ users. Closer inspection reveals that this might be attributable to market manipulation. Specifically, *LooksRare* has its own *ERC-20* tokens to reward users based on the number of trades performed on their platform. This incentivizes fake transactions, where users exchange NFTs frequently simply to earn rewards. This observation inspires us to explore this form of NFT price manipulation in §5.1.

**Collection Price.** A collection is similar to a "brand", consisting of multiple tokens minted from the same smart contract. We next evaluate the value of every token using their last trade price. In total, there are 54,277 (22.23% of the total) NFT smart contracts in the market, and the sum market cap is $20B USD. The majority of NFT collections are surprisingly expensive: the average is $383,660.07 USD. The most expensive collection is an astonishing around $1.4B

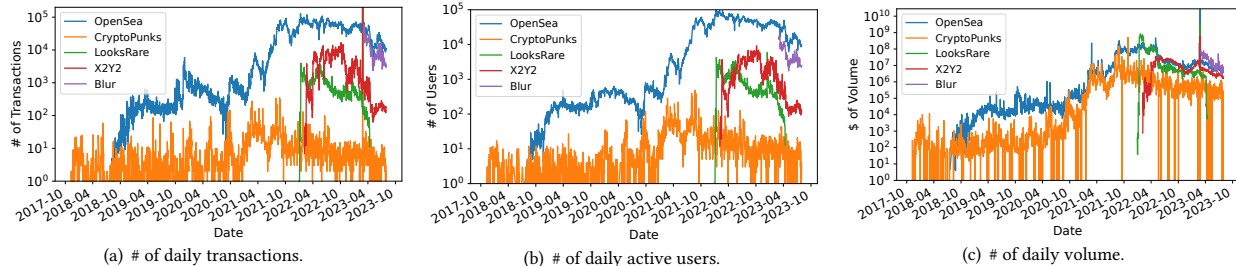

(a) # of daily transactions.

(b) # of daily active users.

(c) # of daily volume.

Figure 4: A comparison of five marketplaces.

USD, with an average of $0.172B USD among the top 100 collections. Only 15.99% collections have a price under $10 USD. We observe a notable set of middle-priced NFTs though: 37.62% exceed $1000 USD. Thus, although many people may think that digital collections seldom have market value, these results suggest otherwise. For context, Table 6 in the Appendix B summarizes the top *collections* that have a value over $700M USD.

However, we find suspicious behaviors within the top collections. Specifically, *Meebits* is one of the most valuable collections in the NFT ecosystem. By inspecting its transactions, we observe that there are 1,655 trades with a price over $1M USD, and 152 trades with a price over $10M USD. Intuitively, these prices are suspiciously high and closer inspection reveals that they are in fact traded by the same small group of users, which also drives us into §5.1.

**User Wealth.** We finally inspect the overall wealth of *users*. We treat the last trade price of each NFT as its value. We identify 1,989,109 *accounts* (users) who hold NFTs. Table 7 in the Appendix shows the top *users* who hold a value of over $10^8$ USD. We identify four addresses that have a sum value over $10^9$ USD and they hold the wealth of over $1.48B USD, which is 7.4% of the total. The top 10% of the holders hold 86.71% of all NFT wealth, with a value of $18B USD. This suggests we are witnessing a consolidation of wealth in the hands of a small minority.

It is difficult to identify *who* these accounts are, however, we do find evidence that some are not authentic. For example, the top user 0xa9 [20] bought 21 tokens in LooksRare whose price is more than $1,000,000 USD during Jan 20th – Feb 10th, 2022. These NFTs belong to the first top collection Meebits, and the third top collection Loot. We conjecture that this is a suspicious activity. We therefore check the trade and find the seller is 0x35 [19], who is also listed in Table 7 of the Appendix B. During the same period, 0x35 simultaneously sells tokens to 0xa9 with a price of more than $10M USD. We find that these accounts buy each others' tokens at a high price, artificially inflating their listed value, which is assumed as a kind of price manipulation and will be discussed further in §5.1.

We have also observed that certain users engage in a large number of trade activities within secondary markets. However, the amount of wealth these users possess remains remarkably small. For instance, 0xc3 [18], who has executed 87,055 trades in secondary markets, yet the wallet still does not hold any value. We are particularly curious about this type of phenomenon, which motivates us to explore further in §5.2.

---

**Summary of NFT Measurement** *The NFT ecosystem became popular in the middle of 2021, with significant and fluctuating growth since then. Dominant projects and NFT holders can trigger huge fluctuations in the NFT ecosystem.*

---

## 5 NFT MARKET MANIPULATION

The previous section has identified preliminary evidence of two kinds of market manipulation [12]. We next deep dive into two types of market manipulation: (*i*) wash trading, and (*ii*) NFT arbitrage.

### 5.1 NFT Wash Trading

Wash trading occurs when a set of accounts buy and sell the same assets multiple times in a short period, to deceive other (normal) market participants about an asset's price.

**Pilot Study.** Our prior analysis of NFT markets (see §4.3) provides evidence of this type of malicious behavior (e.g., market rules of *LooksRare* and fake trades of *Meebits*). This motivates us to conduct a pilot study. To inspect the patterns of wash trading, we define a *seller, buyer* pair, which can be represented as a triplet: *<seller, buyer, weight>*. Because the *Meebits* NFTs are sold in the *LooksRare* marketplace, we select the top 50 seller-buyer pairs according to their sum trade frequency, and represent them in a chord diagram, as shown in Fig. 5. The different color blocks represent different buyers and sellers; and the width of the arrows represents the trade frequency. There are clearly seller-buyer pairs who exchange a large number of NFTs. We also find a non-negligible number of exchanges where the two-way flow of assets are very similar — a classic sign of wash trading. Via manual inspection, we confidently identify 31 users who are almost certainly performing wash trading. From this, we identify three kinds of patterns (motifs), as shown in Fig. 6. *Motif 1:* Wash trading can happen between two users, whereby they buy and sell tokens with each other. *Motif 2:* Wash trading can happen between many pairs of accounts, with a single central user. *Motif 3:* Wash trading can occur in a cycle (i.e., a minimum of three users).

**Detection Approach.** The current methods for detecting wash trading [29, 35, 37, 40, 43, 44, 47] rely on a few basic patterns. However, we aim to design a novel way to identify more wash trading patterns. Specifically, our method estimates the minimum number of wash trading bots and then makes an effort to filter out cases where bots are not involved in wash trading. Based on the observation, we design an automated approach to uncover the wash trading

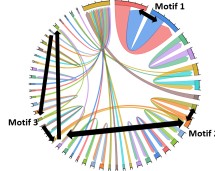
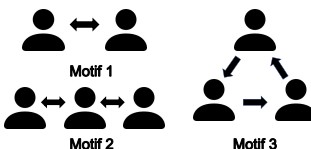

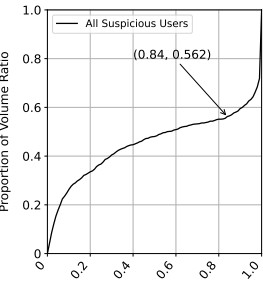
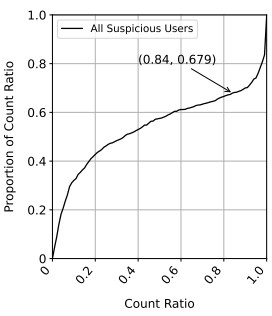

**Figure 5: The top-50 pairs of LooksRare.**

**Figure 6: Three kinds of wash trading motifs.**

**Figure 7: CDF of *volume ratios* for per user.**

**Figure 8: CDF of *count ratios* for per user.**

activities in the ecosystem. Note that the three kinds of motifs we identified in the pilot study are the most simple ones that may not cover all sophisticated wash trading behaviors in the wild. Thus, in contrast to existing works that rely solely on the summarized patterns of wash trading activities, we seek to uncover the wash traders behind them, and then reveal their diverse wash trading behaviors.

To achieve this, we first apply a general heuristic method to flag suspicious trading activities, based on which we label the bots that perform wash trading Based on the wash trading patterns observed in the manually identified bots, we further cluster the trades performed by the bots. Thus, we distinguish normal trades from potentially malicious ones. Specifically, our approach can be divided into four steps.

**Step 1: Selecting Suspicious Trading Pairs.** We first define a triple <*seller, buyer, weight*>, where the weight represents the number of trades between two users. Next, we filter any pairs whose sellers or users are the *official address*. For the remaining pairs, we observe that 98.75% have under five trades, which we did not observe abnormal behaviors by sampling 100 such pairs. Thus, we inspect the remaining pairs that have at least five trades as suspicious ones. We notice that all the wash trading pairs trade intensely during certain short periods (usually within 1 day). Thus, we extract all that have performed their trades within a 48 hour time window. *Step 1 finds 482,274 suspicious trading pairs.*

**Step 2: Heuristic Detection.** Based on the above pairs, we search for all cases of the three *Motifs* discussed in the pilot study. Note that this step may involve false positives, however, the issue will be alleviated in our following step.

*Motif 1: Wash trading between two users.* The first motif is where two users exchange NFTs directly between each other. To detect these from our suspicious set of users, we first compute the volume of reciprocal trades between each pair. This is modeled as a quintuple: <*user1, user2, to weight, from weight*>. If the user pairs are wash trading, the balance of trade between the two users should be approximately *equal*. Thus, we exclude any pairs where there is a over 10% difference between the incoming/outgoing trade flow. The remaining set are assumed to be wash traders.[1]

*Motif 2: Wash trading with central users.* The second motif is where a central user trades with many other users, as shown in Fig. 6. Each individual trade therefore looks similar to *Motif 1*, with a single central high degree user. In fact, these are the users who

appear many times in the results of our *Motif 1* analysis. Thus, we identify *Motif 2* users by extracting all users identified more than once in *Motif 1*.

*Motif 3: Wash trading cycle.* The third form of wash trading is a cycle, containing at least three nodes. To extract all such cases, we generate a directed graph of sellers and buyers using the marketplace dataset. We then extract all the simple cycles that exist within the suspicious pairs, described in *Step 1*. Also, for the same reason, we calculate all the simple cycles in the directed graph and again filter any where the absolute differential value of trade frequency between each pair is over 10%.

*Step 2 flags 454,537 suspicious trades according to the three trading motifs from 246,295 trading pairs in Step 1, associated with 15,148 users.*

**Step 3: Labelling Wash Trading Bots.** The previous step is quite straightforward, yet it may contain false positives (based on a fixed threshold), and it may not cover the advanced tactics used by wash traders. Thus, we next seek to identify the wash trading bots accurately from the result of *Step 2*, and further expand our *wash trading motifs* by analyzing all their trading activities the motifs did not cover.

Specifically, we introduce two metrics to label the bots. For this, we sum up the *wash trading volume* from the 15,148 users detected in *Stage 2*. Then, we sum up the *total trading volume* of each of those users (using the marketplace dataset). After that, we calculate the ratio between those two numbers for every user, termed the *volume ratio*. Similarly, we calculate the ratio between the *wash trading count* and *total trading count* as *count ratio*. We argue that, since *wash trading bots* primarily perform wash trading, they should have either the *volume ratio* or *count ratio* near 1. If *either of* the two ratios are over a certain threshold for a specific user, we assume the user is a wash trading bot and all the trades performed by that specific account are wash trading. We next try to determine a suitable threshold. Fig. 7 and Fig. 8 are the cumulative distribution graphs of *volume ratio* and *count ratio*. The curves for both graphs increase slowly while *volume ratio* or *count ratio* is around 0.5, and rapidly increase as the *volume ratio* or *count ratio* nears 1. After 0.84 for *volume ratio* and *count ratio*, the curves increase rapidly, indicating any user above this threshold has a high possibility to be a wash trading bot. Note, the threshold is not 100%, as these bots are confirmed to have other kinds of wash trading behaviors.

Thus, we heuristically set the thresholds as 0.84 for *volume ratio* and *count ratio*. Among the 15,148 suspicious users detected in *Step*

---

[1]Note, the results of *Motif 1* naturally overlap with *Motif 2*. The results from *Motif 2* is the subset of those from *Motif 1*. We therefore remove the results from *Motif 2* and retain them for *Motif 1*.

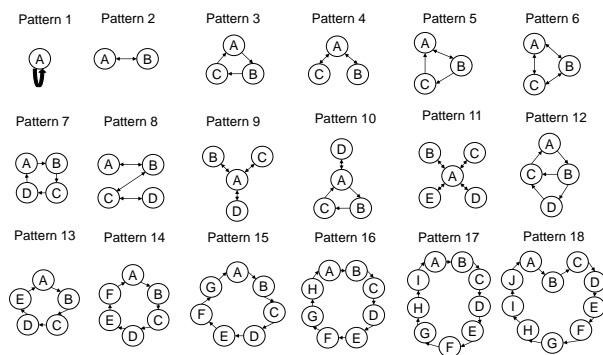

**Figure 9: Summary of wash trading patterns.**

*2*, we therefore label 826 bots as wash trading bots. Even if we make slight adjustments to the thresholds for *volume ratio* or *count ratio*, the identification of users in this step remains relatively consistent, which validates our choice of thresholds. To validate our heuristics, we manually check 100 of the 826 bots by sampling their trade activities, and confirm that they are all wash trading bots, which can ensure that we can get a lower-bound analysis of the issue. *Step 3 finds 826 bots from 15,148 users labelled in Step 2, flagging 85,516 suspicious trades with $24,876,390,650.34 USD trading volume.*

**Step 4: Clustering.** Note that not all trades carried out by a bot necessarily involve wash trading. To filter out transactions that are not related to wash trading, we rely on the identified wash trading patterns within these bots and group together the wash trading activities within them. Consequently, we proceed to expand the trading patterns of both the trades found in *Step 2* and the newly flagged trades. This results in a comprehensive representation of wash trading behaviors, as illustrated in Fig. 9. The types of discovered patterns are beyond the scope of previous research [29, 35, 37, 40, 43, 44, 47], underscoring the effectiveness of our approach in uncovering new patterns. Based on the discovered patterns, we proceed with clustering to identify and exclude non-malicious trades. *Step 4 flags 60,971 wash trades from 85,516 trades labelled in Step 3, with $24,775,694,029.02 USD trading volume performed by 826 bots.*

**Results.** We flag 60,971 wash trades performed by 826 bots. These actions constitute a remarkable $24,775,694,029.02 USD, which means that at least 23.03% of NFT activity on secondary market is created by wash trading. Table 2 summarizes the breakdown of wash trading across all five marketplaces, and presents the top-8 NFT collections that have the largest wash trading volume.

Blur, as a marketplace that get popular in 2023, also have wash trading. Therefore, wash trading is a consistent problem within the NFT ecosystem. There are also notable differences across the marketplaces. Both *CryptoPunks* and *OpenSea* have only a few wash traders, whereas the vast majority takes place on *LooksRare* (over $22B USD). To explain this, we turn to the *LooksRare* official documentation [38]: "*all collections now generate trading rewards. No minimum volume required - you earn LOOKS every time your buy or sell an NFT on LooksRare, from any collection!*". This is a likely explanation for the large volume of wash trading, as users only pay a small trade fee to gain LOOKS token as rewards. This mirrors our

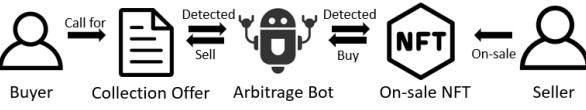

**Figure 10: The flow of NFT arbitrage.**

prior observation, showing wash trading is common in *LooksRare*: From the 122 collections, exhibit 20,945 wash trading behaviors with over $22B USD fake history trading volume.

## 5.2 NFT Collection Offer Arbitrage

Cyclic arbitrage of fungible tokens [46] occurs because the exchange rates between different pairs of tokens in *DEXes* are not always perfectly in sync, opening up arbitrage possibilities for cyclic trading. In some countries, digital arbitrage activities may be regulated or restricted, particularly in financial markets such as currency or stock trading.[2] We therefore conjecture that arbitrage might also happen in the NFT ecosystem. Here, we refer to cycle arbitrage in traditional cryptocurrencies as *traditional e-arbitrage*, and arbitrage in the NFT ecosystem as *NFT-arbitrage*.

**Overview of NFT-Arbitrage.** Compared to *traditional e-arbitrage*, the *unique* characteristics of NFTs open up the possibility of arbitrage in a different way. Figure 10 shows the general process of NFT-arbitrage. Unlike traditional e-arbitrage, arbitrage of NFTs always begins with a *collection offer*. A collection offer is like a "wanted" for any NFT in a specific collection. In *OpenSea*, *WETH (Wrapped Ether)* is needed to make a collection offer. After raising the offer, it is shown in the OpenSea official website and the user needs to wait for the echo. *X2Y2* and *LooksRare* also have approximately the same process. To successfully perform NFT-arbitrages, three conditions must be met: (*i*) A collection offer must be raised by someone else; (*ii*) An NFT from a target collection must be listed for sale on the market; and (*iii*) The output (collection offer price) must outweigh the input (gas fees, handling fee and purchase fee). Arbitrage bots therefore must monitor the collection offers posted on marketplaces. If these three conditions are satisfied, the bot will automatically buy the token listed on the market and sell it to the collection offer. Note, to avoid undesirable changes in price, the buy and sell actions must take place within a single smart contract transaction.

**Detection Method.** In NFT arbitrage, the buy-and-sell actions should be completed within one transaction. This inspires us to design an effective detection method. We refer to the trade dataset as $T$. All the users involved in the trades are in set $U$. Every trade in $T$ consists of the seller, buyer and other information. If $T_1$ and $T_2$ match the following five criteria, we label it as arbitrage: (*i*) The two trades happen in a single transaction, i.e., $T_1.transaction\_hash = T_2.transaction\_hash$. (*ii*) The token of the trade is the same, i.e., $T_1. < contract\_address, token\_id > = T_2. < contract\_address, token\_id >$. (*iii*) If the type of the token is *ERC-1155*, the amount of tokens in these two trade should be the same, i.e., $T_1.amount = T_2.amount$. (*iv*) The price of the first trade should be less than the second one, i.e., $T_1.price < T_2.price$. (*v*) To avoid including

---

[2]We consider this a type of market manipulation. However, there are differing opinions on to what extent this constitutes market manipulation vs. strategic trading.

**Table 2: Summary of wash trades we identified. The column "$ of Wash Trades" is the total history volume generated from wash trades. The column "$ of All Trades" is the total history volume generated from all the trades.**

| | Name or Address | # of Wash trades | $ of Wash trades | $ of All trades | % of Fake history volumn |
|---|---|---|---|---|---|
| | LooksRare | 20,945 | 22,230,486,364.41 | 31,473,916,119.27 | 70.63% |
| | X2Y2 | 11,765 | 2,059,696,277.77 | 5,920,282,010.60 | 34.79% |
| Marketplace | OpenSea | 22,766 | 453,034,260.52 | 64,231,558,049.82 | 0.71% |
| | Blur | 5,489 | 31,187,981.66 | 3,219,154,421.63 | 0.97% |
| | CryptoPunks | 6 | 1,289,144.65 | 2,702,620,665.80 | 0.04% |
| | Terraforms (TERRAFORMS) | 10,884 | 11,674,819,866.45 | 12,320,656,847.36 | 94.75% |
| | Meebits | 7,720 | 7,071,806,358.50 | 10,061,077,548.79 | 70.29% |
| | dotdotdot (dotdotdot) | 1,727 | 1,838,298,518.38 | 2,724,498,012.57 | 67.47% |
| Collection | More Loot (MLOOT) | 1361 | 1,451,415,137.95 | 4,880,660,670.93 | 29.73% |
| | Loot (LOOT) | 616 | 600,663,668.40 | 1,009,972,739.01 | 59.47% |
| | Audioglyphs (AG) | 738 | 377,160,076.54 | 380,729,286.42 | 99.06% |
| | CATGIRL ACADEMIA (CAT) | 422 | 339,172,692.24 | 339,515,284.64 | 99.85% |
| | CryptoPhunksV2 (PHUNK) | 125 | 275,645,653.57 | 285,390,139.01 | 96.56% |

false positives by wash trading (see §5.1), $T_1.seller! = T_2.buyer$ and $T_1.seller! = T_1.buyer$ and $T_2.seller! = T_2.buyer$ If all five criteria are fulfilled, we regard this trade (pair) as arbitrage.

**Results.** Through the above methodology, we identify 629 users who exhibit arbitrage behavior. These users perform 157,302 cases of arbitrage. We define the *arbitrage profit* as the sale price minus the the bot purchasing price; and the *arbitrage volume* as the price that the sale price plus the bot purchasing price. These arbitrages sum up to a profit of $25,310,982.22 USD and a volume of $186,188,047.24 USD. There are 38,819 cases of cross-marketplace arbitrage and 118,483 times of same-marketplace arbitrage. Table 8 (in Appendix) summarizes the top-5 arbitrage bots, each of which has gained a profit of over $800K USD. That said, 80.4% of the bots perform arbitrage fewer than 20 times, indicating that a small set of bots gain the majority of profits via arbitrage. There are 5,443 collections that have been arbitraged, and the average number of cases per collection is 28.90. Interestingly, we observe that some of the arbitraged collections also appear among the most valuable collections, e.g., *OpenSea Shared Storefront* and *Otherdeed*. This is intuitive because of the demand for NFTs from popular collections, i.e., the more offers are raised, creating more potential for arbitrage.

> **Summary of NFT Market Manipulation** *Wash trading and NFT arbitrage both take place, affecting billions of dollars on market. At least 23% of NFT market trading is fake, generated by 826 bots. 157,302 NFT arbitrage cases are performed by 629 bots, with profits of over $25M USD.*

## 6 RELATED WORK

**Research on NFTs.** Wang et al. [45] study the technical component of NFTs, explaining their design and properties. They also discuss potential security issues. However, their conclusions are based on the design of NFTs or individual cases; they lack an empirical investigation into the overall NFT ecosystem, unlike this paper. Ante et al. [23] study 14 top collections of NFTs, as well as the relationship between NFTs and Ethereum by evaluating the exchange rate and other economic factors. They only focus on several large NFT projects. There are also some researchers who focus on

the usage of NFTs [24–26, 28, 31, 32, 41]. However, none of these works provide a systematic overview of the NFT ecosystem from both an on-chain data and market view.

**Crypto Market Manipulation.** There have been works identifying price manipulation on blockchains. Cong et al. [33] study wash trading with fungible cryptocurrencies. Rug pull schemes have also been detected by Mazorra [39], Xia [48] and Huang [36]. Prior studies explore price manipulation behavior on *Ethereum* or other chains from different angles, such as wash trading [29, 35, 37, 40, 43, 44, 47], crypto rug pulls [39, 48], crypto arbitrage [22, 27, 30, 46]. Our approach can identify more patterns of wash trading than previous work and ensure the lower bound of bots. We also automatically detect the arbitrage within *NFTs*, which is different from existed detection of arbitrage within fungible tokens.

## 7 LIMITATION

Our study carries certain limitations. Addressing these are the foundation of our future work. First, we only track five major NFT markets. Since these markets have a complicated design, manual efforts are still a necessary part, which means we may miss some cases of misbehavior in smaller markets. That said, these markets account for most of the trading volume and will likely reflect most trading (mis)behaviors. Second, our detection for wash trading is simple, and may miss certain cases such as wash trading bots with a low impact. However, we emphasize that we are able to find more patterns than prior works [29, 35, 37, 40, 43, 44, 47].

## 8 CONCLUSION

This paper has conducted the first large-scale analysis of the NFT ecosystem from both an on-chain and market view. Based on datasets of both NFT transactions and trades on major marketplaces, we have looked at various dimensions. We have shown that the ecosystem is subject to substantial market manipulation, and over 23% of NFT market volume is generated artificially. Arbitrage also takes place in NFT ecosystem, bringing over $25M USD profits for arbitrager. Our exploration suggests that the governance of NFTs needs to be improved, and it is urgent for the research community to propose effective countermeasures to address NFT issues.

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

# APPENDIX

# A DETAILS OF COLLECTING SECONDARY MARKET DATASET

First, we manually inspect all the *external functions* or *public functions* in the smart contracts to find functions that directly handle trading-related information. The smart contracts emit an event when the trade process completes. We thus check the event declarations emitted by these contracts, and find several events containing information related to NFT trades. All *official smart contracts* and *relative events* of marketplaces that are taken into consideration are listed in Table 3. To automate the process, we must map the raw data in the logs to useful trading information. Thus, we take the aforementioned external and public functions as the entries of these market smart contracts, and go through the execution path in which an NFT trade can successfully complete and emit the corresponding events. We do this to help understand each field of the logged data in these trading-related events. With this insight, we manually construct a mapping between *trading information* and *on-chain log data* to help us parse the remaining data in the logs. Finally, the extracted trading information consists of the contract address, token id, buyer's address, seller's address, currency address and currency amount. We use *Ethplorer* [7] to obtain the daily average exchange rate (to USD) of all encountered cryptocurrency tokens. We compile this data for all trades within the four marketplaces.

# B EXTREME CASES IN NFT ECOSYSTEM

As discussed in §4.1, we list the top in-degree accounts in Table 4, top out-degree accounts in Table 5, the most valuable collections in Table 6, the wealthiest users in Table 7. The top arbitrage bots that perform arbitrage with a profit of over $800K.

**Table 3: Smart contracts and addresses about the top-five NFT secondary markets.**

| | Relative Segment Name | Relative Address |
|---|---|---|
| OpenSea | Seaport Address (V1) | 0x00000000006c72100d161c57ada5bb2be1ca79 |
| | Seaport Address (V2) | 0x00000000006c3852cbef3e08e8df289169ede581 |
| | Seaport Address (V2) | 0x00000000006c3852cbef3e08e8df289169ede581 |
| | Seaport Address (V3) | 0x00000000000006c7676171937c444f6bde3d6282 |
| | Seaport Address (V4) | 0x0000000000000ad24e80fd803c6ac37206a45f15 |
| | Seaport Address (V5) | 0x00000000000001ad428e4906ae43d8f9852d0dd6 |
| | Seaport Address (V6) | 0x0000000000000adc04c56bf30ac9d3c0aaf14dc |
| | Wywern Address (V1) | 0x7be8076f4ea4a4ad08075c2508e481d6c946d12b |
| | Wywern Address (V2) | 0x7f268357a8c2552623316e2562d90e642bb538e5 |
| X2Y2LooksRare | LooksRare Address | 0x59728544b08ab483533076417fbbb2fd0b17ce3a |
| | TakerAsk Event | 0x68cd251d4d267c6e2034ff0088b990352b97b2002c0476587d0c4da889c11330 |
| | TakerBid Event | 0x95fb6205e23ff6bda16a2d1dba56b9ad7c783f67c96fa149785052f47696f2be |
| | X2Y2 Address | 0x74312363e45dcaba76c59ec49a7aa8a65a67eed3 |
| | Inventory Event | 0x3cbb63f144840e5b1b0a38a7c19211d2e89de4d7c5faf8b2d3c1776c302d1d33 |
| | Profit Event | 0xe2c49856b032c255ae7e325d18109bc4e22a2804e2e49a017ec0f59f19cd447b |
| Blur | Blur Marketplace 1 | 0x000000000000ad05ccc4f10045630fb830b95127 |
| | Blur Marketplace 2 | 0x39da41747a83aee658334415666f3ef92dd0d541 |
| | Blur Marketplace 3 | 0xb2ecfe4e4d61f8790bbb9de2d1259b9e2410cea5 |
| CryptoPunks | CryptoPunks Address | 0x47e3cd837ddf8e4c57f05d70ab865de6e193bbb |
| | PunkBought Event | 0x58e5d5a525e3b40bc15abaa38b5882678db1ee68befd2f60bafe3a7fd06db9e3 |

**Table 4: Top five indegree accounts.**

| Account address | Indegree | Identity |
|---|---|---|
| 0x0000000000000000000000000000000000000000 | 3,462,665 | Official Account |
| 0x283af0b28c62c092c9727f1ee09c02ca627eb7f5 | 2,160,818 | ENS |
| 0x000000000000000000000000000000000000dead | 917,025 | Marketplace |
| 0x83c8f28c26bf6aaca652df1dbbe0e1b56f8baba2 | 916,057 | Official Account |
| 0x39da41747a83aee658334415666f3ef92dd0d541 | 746,025 | Marketplace |

**Table 5: Top five outdegree accounts.**

| Account address | Outdegree | Identity |
|---|---|---|
| 0x0000000000000000000000000000000000000000 | 148,500,667 | Official account |
| 0x283af0b28c62c092c9727f1ee09c02ca627eb7f5 | 2,160,811 | Ethereum Name Service (ENS) |
| 0x83c8f28c26bf6aaca652df1dbbe0e1b56f8baba2 | 915,920 | Marketplace |
| 0x39da41747a83aee658334415666f3ef92dd0d541 | 745,931 | Marketplace |
| 0x6109dd117aa5486605fc85e040ab00163a75c662 | 342,806 | ENS: Wallet |

**Table 6: Top five collections that have largest value.**

| Collection address | Value | Name |
|---|---|---|
| 0xb47e3cd837ddf8e4c57f05d70ab865de6e193bbb | 1,434,932,716.61 | CRYPTOPUNKS |
| 0xbc4ca0eda7647a8ab7c2061c2e118a18a936f13d | 1,237,039,866.62 | BoredApeYachtClub |
| 0x7bd29408f11d2bfc23c34f18275bbf23bb716bc7 | 723,464,798.77 | Meebits |
| 0x495f947276749ce646f68ac8c248420045cb7b5e | 722,008,516.34 | OpenSea Shared Storefront |
| 0xa7d8d9ef8d8ce8992df33d8b8cf4aebabd5bd270 | 717,895,694.75 | Art Blocks |

## C   ADVICE FOR COMMUNITY

Our findings are of key importance to the stakeholders in the NFT community. (*i*) *The governance of the NFTs:* The ecosystem has witnessed significant growth. However, considering that market manipulation and security issues are prevalent, the governance of NFTs needs to be improved. We believe a platform for evaluating NFT tokens is needed to mitigate the impact of market manipulation and security issues. The platform can adopt techniques in this work for monitoring the trades and contracts to identify wash trading, arbitrage or other security issues. Our detection techniques can be further embedded in services like markets and wallets and act as reminders for investors when they try to interact with potential high-risk NFTs. (*ii*) *NFT Creators:* The official NFT creators should be aware of potential market manipulation. It is their responsibility to actively search, understand and identify these risks. After the

**Table 7: Top users that hold the largest value of NFTs.**

| User account address | Total value(USD) |
|---|---|
| 0xa99a76dddbb9678bc33f39919bc76d279c680c89 | 592,586,076.20 |
| 0x9b5a5c5800c91af9c965b3bf06ad29caa6d00f9b | 511,029,067.58 |
| 0x73ec85489681da69fb52d8b25aee0091eb2925ce | 211,809,146.96 |
| 0x83c8f28c26bf6aaca652df1dbbe0e1b56f8baba2 | 165,675,922.31 |
| 0x35d0ca92152d1fea18240d6c67c2adfe0cca287c | 46,622,000.73 |

**Table 8: Top-5 bots that perform arbitrage with a profit of over $800K.**

| Bot address | # of Arbitrage times | $ of Arbitrage profits | $ of Arbitrage volume |
|---|---|---|---|
| 0x8f44e22ac221cc25a46289d1c307d4f34a4dd6c2 | 9,248 | 5,741,249.36 | 9,253,846.63 |
| 0x9e9346e082d445f08fab1758984a31648c89241a | 1566 | 2,114,383.24 | 7,789,423.87 |
| 0x553eea17185e5ae6bb72f9528a4c3fc1a844b859 | 986 | 1,268,150.30 | 6,485,947.10 |
| 0xc34349fbedd527215aae19b2e4626254ec29a13d | 43,446 | 1,262,516.60 | 68,810,679.29 |
| 0x6b58007b960016b2f559dbfd809ac4dcb1febdfe | 717 | 821,175.15 | 4,000,063.41 |

launch of their projects, they should regularly publish security bulletins to remind users. (*iii*) *Investors:* For NFT investors, the awareness of potential risks on NFTs should be improved. Rather than just searching for high-value or over-hyped NFTs, they should rely on trusted sources to investigate the trading history of their potential purchases. They also need to perform research on the developers behind the projects to check whether they have a bad reputation in prior projects.

