# OpenReview forum: "Unveiling the Paradox of NFT Prosperity"
_ACM.org/TheWebConf/2024/Conference — TheWebConf24_

### Official Review · Reviewer_e239 · 2023-11-15

**Novelty:** 5
**Technical Quality:** 5

**Review:**

The paper collects and analyzes a large dataset of NFT trades and transfers, the first such large-scale study. Data is collected from the Ethereum ledger and five major NFT marketplaces. Two kinds of selfish behavior is analyzed: 1) Wash trading, in which an NFT is made to appear desirable by trading it repeatedly, and 2) arbitrage, in which pricing inconsistencies are exploited to make quick profits.

Both kinds of behavior are found to be widespread (making up 23% of market activity). For wash trading, the types of patterns detected go beyond what has been possible in previous works.

This seems like a solid work to me. One thing that is unclear to me to what extent we would expect NFTs are fundamentally different than other cryptographic tokens such as coins (see question below).

**Questions:**

- Why would one expect NFT trading to be different from trading with cryptocurrencies? It seems to me that they are abstractly the same, speculative assets, a bit like distinguishing between dollar bills with and without pictures.

- Would you consider a different title? It seems to me that "Unveiling the Paradox of NFT Prosperity" does not accurately capture the content of the paper

- Do you plan to make your dataset available to make it easier for others to build on your study?

**Reviewer Confidence:**

2: The reviewer is willing to defend the evaluation, but it is likely that the reviewer did not understand parts of the paper

**Scope:**

4: The work is relevant to the Web and to the track, and is of broad interest to the community

---

### Official Review · Reviewer_UMJi · 2023-11-24

**Novelty:** 6
**Technical Quality:** 5

**Review:**

Summary: The paper presents a measurement study of the NFT ecosystem, first by providing a longitudinal overview of the evolution of the ecosystem, and second of malicious trading patterns on NFT markets that might amount to market manipulation.

Reasons to accept:
* longitudinal measurement of NFT on-chain transfers and in-market trading
* measurement of NFT "wash trading" exposing suspicious behavior

Reasons to reject:
* the impact/relevance of the findings/takeaways from the longitudinal ecosystem measurement is often unclear, making that part of the paper seem a bit superficial
* little in terms of validation of the "wash trading" detection methodology
* important methodology detail can only be found in the appendix but should be part of the main paper

The paper is reasonably easy to follow, covers a timely topic, and presents interesting findings overall. The main findings appear sound, but there is potential for improvement. My main concern is that the methodology is often vague, and key details (such as how token values were converted to USD) are either relegated to the appendix, or omitted entirely. I would recommend the authors integrate Appendix A into the main paper. Section 5 needs more detail in the methodology (e.g., for Step 4, I was wondering whether it used the same or a different method to identify patterns, and how the clustering was done). Section 5 could also benefit from more validation, for example for the selection of the threshold/wash trading detection. For example, the authors could sample and manually annotate not only users declared wash bots (above the threshold), but also users close to (but below) the wash trading threshold. Alternatively, the authors could highlight from the beginning that they only aim to find a lower bound; they do mention this later but it is helpful to be upfront about the goals and limitations.

Section 4 appears to be more polished than Section 5, and I appreciate that the authors tried to highlight the implications of their findings and provide a summary with the key takeaways. Unfortunately, some of the mentioned implications appear trivial or unfinished/only partially explained. For example, when the authors state that "The top 10% of contracts mint 90.57% of all tokens. This raises serious concerns about the the true level of decentralization in the ecosystem, as the removal of a small number of stakeholders would remove the majority of “creativity”." or find "significant centralization in production of NFTs", it is unclear whether this really is (or should be) a concern: Is there a risk that one of these stakeholders is removed? Would another stakeholder not be able to take their place? What are the concrete disadvantages to users from this centralization?

If at all possible, I would recommend the authors try to integrate Appendix C into the main paper, perhaps in shortened form. It justifies the usefulness of the authors' work. (Mostly referring to Section 5, thus coming up with something comparable relating to Section 4 would be helpful.)

Minor points:
* End of Section 2.2: 2x citing LooksRare instead of X2Y2
* Unclear to me why it makes no sense to count ERC 1155 tokens (line 214) but then looking at their creation rate in Section 4.1
* Section 4.1: "for the same reason discussed above": briefly restate the reason (many were discussed)
* Section 4.1: define "airdropped"
* Line 460: the total refers to what?
* Line 479: is this conflating "collection" and "NFT"?
* Figure 4c: what is the unit? USD?
* Section 5.2: explain "wrapped Ether"
* Figure 7/8: I agree with the 0.84 threshold for the volume ratio, but why pick the same threshold for the count ratio? It seems the rapid increase starts later and it is unclear why both need to use the same threshold
* Section 5.1: Why calling them motifs first and then patterns?
* Table 2: what is in brackets after the collection name (e.g., "TERRAFORMS")?

**Questions:**

-

**Ethics Review Description:**

no ethical issues

**Reviewer Confidence:**

3: The reviewer is confident but not certain that the evaluation is correct

**Scope:**

4: The work is relevant to the Web and to the track, and is of broad interest to the community

---

### Official Review · Reviewer_G5o5 · 2023-11-24

**Novelty:** 4
**Technical Quality:** 4

**Review:**

The paper examines the evolution of the NFT ecosystem since 2017, conducting a thorough analysis of NFT events and marketplaces through exploratory measurements. The authors compiled a dataset of NFT-related transactions on Ethereum and data from five prominent secondary NFT marketplaces.

I think the paper did a good job of sketching the ecosystem. There are some strengths:
1. The empirical measurements offer valuable insights.
2. The choice of datasets is appropriate, sourced directly from the blockchain, which is likely to be complete.
3. The paper is generally easy to follow, although some figures were referenced before being presented.

However, my primary concern is the novelty of the methods. While the findings are somewhat informative and interesting, most are not surprising; they lack something really novel. For example, the rise of the NFT ecosystem in mid-2021 and the centralisation of NFT holders are not very surprising. The most interesting part is probably the wash trading pattern they discovered, with commendable efforts made for verification.

**Questions:**

1. The secondary markets offer NFT-collateralised loans, so how does it compare to the rest of the transactions?
2. The social network in 4.2 can be more comprehensive beyond degree distributions e.g., how do the top participants interact with each other over time and how do they build up their relationship (if any)?
3. Figure 1 is referenced on page 3 but appears later on page 4. This inconsistency makes it harder to follow.
4. Appendix C should be moved to the main body.

**Reviewer Confidence:**

3: The reviewer is confident but not certain that the evaluation is correct

**Scope:**

4: The work is relevant to the Web and to the track, and is of broad interest to the community

---

### Official Review · Reviewer_zobK · 2023-11-28

**Novelty:** 5
**Technical Quality:** 5

**Review:**

The current work focuses on performing a comprehensive study of the NFT ecosystem. For this, authors claim to have gathered the largest NFT dataset so far, and analyse the same to determine possibility of market manipulation. Specifically, they measure the prevalence of wash trading and arbitrage. The authors analyse smart contracts of five major NFT markets, which covers over 98% of the total trade volume over Ethereum. The authors provide a systematic study that investigates– the different types of NFT events, distribution of participants that drive the ecosystem, a comparative study of the five markets considered, identifying market manipulation via wash trading and arbitrage.

The authors clearly state the various aspects of the ecosystem that they wish to analyse and present the findings intelligibly. Since the authors claim that such a comprehensive study is being carried out for the first time for NFTs, the work plays a key role in unravelling the various aspects of the ecosystem. Although the current work makes several observations regarding the current practices in NFT markets, it is unclear how these findings would aid in the improving the NFT ecosystem.

**Questions:**

Did the study lead to having any insights on strategies that could be effective as countermeasures?

**Reviewer Confidence:**

3: The reviewer is confident but not certain that the evaluation is correct

**Scope:**

3: The work is somewhat relevant to the Web and to the track, and is of narrow interest to a sub-community

---

### Official Review · Reviewer_79rv · 2023-11-28

**Novelty:** 4
**Technical Quality:** 4

**Review:**

The research content of this paper is completely beyond my research field, so I cannot give an accurate evaluation or comment.

I will refer to the opinions of other reviewers to determine my final rating.

**Questions:**

N/A

**Reviewer Confidence:**

1: The reviewer's evaluation is an educated guess

**Scope:**

3: The work is somewhat relevant to the Web and to the track, and is of narrow interest to a sub-community

---

### Official Review · Reviewer_KTgZ · 2023-12-01

**Novelty:** 2
**Technical Quality:** 3

**Review:**

The paper examines wash trading and arbitrage across five NFT marketplaces by performing analysis on 242M transfer logs.  The major problem I have with this paper is that the methodology as presented is very haphazard, and it is not clear what the technical contributions of the work are compared to previous work.
 For example, the authors mention 7 previous works on wash trading detection and then claim that their methodology is novel without really comparing against those works or providing discussions. Furthermore, wash trading has been previously explored. E.g., https://dune.com/hildobby/nfts-wash-trading, and it is not clear how these results provide any additional insights compared to those. Many design choices have been made without proper analysis of the logic. E.g., all wash trading pairs trade intensely during certain short periods; thus, we chose a 48-hour time window.
The arbitrage detection method is similarly trivial and the paper does not yield significant insights or discussions beyond the obvious first take.

**Questions:**

- How does your wash trade detection method differ from the previous work?

**Reviewer Confidence:**

3: The reviewer is confident but not certain that the evaluation is correct

**Scope:**

3: The work is somewhat relevant to the Web and to the track, and is of narrow interest to a sub-community

---

### Decision · Program_Chairs · 2024-01-22

**Decision:**

Accept

**Comment:**

Strengths:
 + Useful empirical measurements
 + Nice longitudinal analysis, over time
 + Identifies potentially problematic "wash trading"
 + Easy to follow (modulo some figure issues)
 + Comprehensive datasets
 + First comprehensive look at the NFT ecosystem


 Weaknesses:
 - Some methods used in the paper are not well-defined
 - Technical contributions unclear
 - Lack of novelty
 - Many of the findings are unsurprising or do not offer new insights
 - Some of the analysis seems a bit superficial
 - Little validation presented to support the methodology used to identify wash trading

 Recommendation: The study is first of its kind, but lacks technical depth.